# Gut acquisition of *Extended-spectrum β-lactamases-producing Klebsiella pneumoniae* in preterm neonates: Critical role of enteral feeding, and endotracheal tubes in the neonatal intensive care unit (NICU)

**Benboubker Moussa**[iD][1]*, **Bouchra Oumokhtar**[1], **Btissam Arhoune**[2], **Abdelhamid Massik**[2], **Samira Elfakir**[3], **Mohamed Khalis**[4], **Hammad Soudi**[iD][3], **Fouzia Hmami**[1,5]

**1** Faculty of Medicine and Pharmacy, Human Pathology Biomedicine and Environment Laboratory, Sidi Mohammed Ben Abdellah University, Fez, Morocco, **2** Faculty of Medicine and Pharmacy, Microbiology and Molecular Biology Laboratory, Sidi Mohammed Ben Abdellah University, Fez, Morocco, **3** Faculty of Medicine and Pharmacy, Departement of Epidemiology and Public Health, Sidi Mohammed Ben Abdellah University, Fez, Morocco, **4** International School of Public Health, Mohammed VI University of Health Sciences, Casablanca, Morocco, **5** Neonatal Intensive Care Unit, University Hospital Hassan II, Fez, Morocco

* moussa.benboubker@usmba.ac.ma

## Abstract

### Background

Klebsiella spp. can colonize the intestine of preterm neonates, and over-growth has been associated with necrotizing enterocolitis, hospital-acquired infections, and late-onset sepsis. This could lead us to suggest that the clinical pertinence of intestinal colonization with *ESBL* in preterm neonates appears to be important. We conducted this study to characterize the genetic proprieties *of ESBL-producing Klebsiella pneumoniae (ESBL-KP)* under clinical isolates and to describe the risk factors for the intestinal tract acquisition event during hospitalization.

### Methods

One hundred and thirteen premature infants were recruited from the neonatal intensive care unit (NICU). All newborns are issued from the birth suites of the pregnancy department. Two rectal swabs were planned to define *K. Pneumoniae* intestinal carriage status. ESBL-KP was confirmed by Brilliance *ESBL* selective chromogenic Agar. Antimicrobial susceptibility testing including phenotypic testing and genotypic detection of the most commonly described ESBL genes was done. Logistic regression models were performed to find the variables associated with the acquisition event of *ESBL-KP*.

### Results

A total of 62 (54.86%) premature neonates were colonized with *ESBL-KP*. The rate of *blaSHV, blaTEM, blaCTX-M1, blaCTX-M2, blaCTX-M9, and blaOXA-48* genes among the

**Data Availability Statement:** All relevant data are within the paper and its Supporting information files.

**Funding:** The authors received no specific funding for this work.

**Competing interests:** The authors have declared that no competing interests exist.

isolates was 82, 48, 93.5, 4.8, 11.2 and 3.22%, respectively. We found that *ESBLs K. Pneumoniae* isolates were 100% resistant to *amoxicillin*, *clavulanic acid-amoxicillin*, *cefotaxime*, *ceftazidime*, and *gentamicin*. The regression model is for a given significant association between the tract intestinal of *ESBL-KP* acquisition events and the use of enteral tube feeding (OR = 38.46, 95% CI: 7.86–188.20, p-Value: 0.001), and endotracheal tubes (OR = 4.86, 95% CI: 1.37–17.19, p-Value 0.014).

## Conclusion

Our finding supposes that the enteral feeding tube and endotracheal tube might have a critical role in colonizing the intestinal tract of preterm infants. This highlights the current status of both practices that will require updated procedures in the NICU.

## Introduction

*Klebsiella Pneumoniae* healthcare-associated infections are a serious problem for neonate infants hospitalized in neonatal intensive care units NICU [1]. This pathogen arouses the interest of infection control specialists due to the increasing number of antibiotic-resistant strains. Indeed, more than a third of *Klebsiella. pneumoniae* isolates reported to the European Centers for Disease Prevention and Control were resistant to at least one antimicrobial group, The aminoglycosides are the most common resistance profile with combined resistance to fluoroquinolones and third-generation cephalosporins [2]. Antibiotic-resistance genes can be exchanged between gram-negative bacteria through horizontal gene transfer. The production of other types of *ESBL*, *CTX-M* like, more selective to cefotaxime than to other broad-spectrum cephalosporins, has been increasingly detected in the last years and is also increasing in *K. pneumoniae* [3–5].

Due to the combination of the Newborn's immature immune system's decreased functioning and frequent antibiotic exposure, hospitalized neonates are particularly susceptible to developing *k.pneumoniea* infections [6]. Moreover, it's well known that the appearance of late infections in preterm infants is strongly associated with intestinal carriage, and colonized patients act as a reservoir for the spread of resistant organisms [7].

The first Bacterial colonization of the neonatal gastrointestinal tract occurs from the mother and the immediate inanimate environment [8,9]. However, some early clinical factors that might disrupt the normal acquisition process of the first microbiota in the neonate during NICU stay have been identified [10,11], such as prematurity (gestational age <37 weeks), length of stay, previous use of antibiotics, absence of breastfeeding, and Nasogastric tube feeding [12–14]. However, in contexts where the resistant bacteria have attained endemic levels, the dynamics of the spread seem more dependent on the environmental factors in the NICUs. This might lead us to suggest that the clinical pertinence of *ESBL* gut colonization of preterm neonates appears to be important, between 12 and 50% of newborns colonized with *ESBL*-producing bacteria have developed a bloodstream infection with positive blood cultures [15,16].

This study aimed to determine the intestinal tract acquisition of *ESBL-KP prevalence* in preterm neonates, describe their genetic properties, and study the main risk factors associated with colonization events during neonatal intensive care.

## Materials and methods

### Study design

This prospective study was conducted at the neonatology and intensive care unit (NICU) department of the University Hospital of Fez (Morocco) during 1 year from February 2019 to July 2020. The setting was a medical and surgical neonatal intensive care unit with 18 beds divided into 2 sectors (9 beds in each); the first sector corresponds to an intensive care unit and the second to a unit for preterm babies. This NICU is the only one in Fez city (the center of Morocco), with approximately 1.5 million inhabitants. Three seniors, 8 physicians, and 6 nurses are assigned to this ward daily.

According to the *WHO* criteria, we selected all premature infants hospitalized in the NICU to be studied for intestinal carriage, and prematurity was defined as any baby born alive before the end of 37 weeks of pregnancy [17]. Therefore, neonates hospitalized during the weekend who have died or output before 48h of hospitalization were excluded. Preterm infants were screened (a) at admission and (b) during hospitalization for *ESBL-KP* intestinal carriage. Only preterm infants who were not positive for a given species on admission were considered at risk of acquiring *ESBL-KP*.

### Ethical consideration

The study protocol was approved by the Ethics Committee of the Faculty of Medicine and Pharmacy, and the HASSAN II University Hospital of Fez of Morocco, all the parents' babies were informed of the conditions related to the study and gave their written, informed consent. Parents may remove their consent at any time during the study.

### Data collection

Information was obtained from the medical information system and classified as follows: Anamnestic data (gestational age, delivery mode, prolonged premature rupture of membrane history (PPROM), postpartum clinical data (gender, birth weight, associated pathology, prematurity), and possibly evolving clinical data (invasive procedures, length of stay, antibiotic exposure, endotracheal tube and type of feeding).

### Sampling and screening

Two rectal swabs were collected from each preterm newborn. The initial sample was performed up to 6 hours from admission to the NICUs and the second one after 5 days of hospitalization. Rectal swab specimens were enriched in nutrient broth BHI (Brain Heart infusion, Oxoid1) at 37˚C for 24h. Then, they were inoculated on *Mac Conkey* agar plates and then incubated at 37˚C for 24h. The identification of Enterobacteriaceae isolates was performed by classical bacteriological techniques.

### Antimicrobial susceptibility testing

The antibiotic resistance patterns of isolates of *K. pneumoniae* were determined by disc diffusion method in agar according to the EUCAST for 14 discs of antibiotics including *amoxicillin* (10 μg); *amoxicillin/clavulanic acid* (20/10 μg); *cephalothin* (30μg); *cefotaxime* (30 μg); *ceftazidime* (30 μg); *ertapenem* (10 μg); *nalidixic acid* (30 μg); *ciprofloxacin* (5 μg); *norfloxacin* (10 μg); *gentamicin* (10 μg); *amikacin* (30 μg); *Fosfomycin* (50 μg); and *cotrimoxazole* (1.25/ 23.75 μg) used the disk diffusion on *Mueller–Hinton* agar (Bio-Rad, Hercules, CA). *ESBL-KP* was confirmed by a selective chromogenic medium for the screening of *Extended Spectrum-Lactamase-producing Enterobacteriaceae* (Brilliance ESBL Agar, Oxoid). The *K. Pneumoniae*

strain ATCC 700603 was used as a quality control antibiogram control strain for ESBL production.

## Molecular analysis

The DNA extraction was performed according to the method described in the previous studies [18]. In addition, an aliquot for 2μL of the supernatant was used as a DNA template for the PCR.

All *ESBL-KP* strains were screened by polymerase chain reaction (PCR) for the following β-lactamase-encoding genes: *blaCTX-M*, phylogenetic lineage groups 1, 2, and 9; *blaTEM*; *blaSHV*; *blaKPC, blaNDM, blaVIM and blaOXA*-48 "Table 1". Amplification reactions were carried out in a 50 μl volume containing 2 μl of DNA template, 2.5 *mM MgCl2*, 0.4 mM of each forward and reverse primer, 100 mM of each *dNTP*, and 2 U of *Taq DNA Polymerase* (Promega, Madison, WI) in PCR buffer performed and provided by the manufacturer. The known *β-lactamase-producing strains E. coli U2A1790 (CTX-M-1), E. coli U2A1799 (CTX-M-9), Salmonella sp. U2A2145 (CTX-M-2), Salmonella sp. U2A1446 (TEM-1 and SHV-12)* were used as positive controls. PCR products were detected on 1.5% agarose gels (FMC BioProducts, Rockland, ME) stained with ethidium bromide and visualized under Ultra Violet light.

## DNA Sequencing

All amplified products obtained were sequenced to confirm their identification. Both strands of the purified amplicons were sequenced on a 3130 1 Genetic Analyzer (Applied Biosystems, Foster City, CA) using the identical primers that were used for PCR amplification. The nucleotide and protein sequences were analyzed using software from the National Center for Biotechnology Information (NCBI) website (http://www.ncbi.nlm.nih.gov).

**Table 1. Primer and parameters cycling for characterizing *ESBL-K. Pneumoniae* strains.**

| Gene | Primer | Primer sequence (5'-3') | Cycling Parameters | References |
|------|--------|------------------------|--------------------|------------|
| $bla_{SHV}$ | OS-5 | CGCCGGGTTATTCTTATTTGTCGC | Denaturation: at 95˚C for 5 min, followed by 30 cycles at 95˚C for 1 min, | [26] |
| | OS-6 | TCTTTCCGATGCCGCCGCCAGTCA | | |
| $bla_{TEM}$ | A-216 | ATAAAATTCTTGAAGACGAAA | Annealing: at 60˚C for 1 min for *CTX-M*; *SHV* or at 52˚C for 1 min for *TEM; KPC, NDM, VIM and OXA-48* at 50 ˚C for $45_S$ | [19] |
| | A-217 | GACAGTTACCAATGCTTAATCA | | |
| $bla_{CTX-M-1}$ | Ctx-$_{M1}$(+) | GGTTAAAAAATCACTGCGTC | Extension: at 72˚C for 1 min, ending with a final extension period of 72˚C for 7 min. | [19,26] |
| | Ctx-$_{M1}$(-) | TTGGTGACGATTTTAGCCGC | | |
| $bla_{CTX-M-2}$ | Ctx-$_{M2}$(+) | ATGATGACTCAGAGCATTCG | | [31,20] |
| | Ctx-$_{M2}$(-) | TGGGTTACGATTTTCGCCGC | | |
| $bla_{CTX-M-9}$ | Ctx $_{M9}$(+) | ATGGTGACAAAGAGAGTGCA | | [19,20] |
| | Ctx-$_{M9}$(-) | CCCTTCGGCGATGATTCTC | | |
| blaNDM | Ndm (+) | GGTTTGGCGATCTGGTTTTC | | [21] |
| | Ndm (-) | CGGAATGGCTCATCACGATC | | |
| blaKPC | Kpc (+) | CGTCTAGTTCTGCTGTCTTG | | |
| | Kpc (-) | CTTGTCATCCTTGTTAGGCG | | |
| blaVIM | Vim (+) | GATGGTGTTTGGTCGCATA | | [22] |
| | Vim (-) | CGAATGCGCAGCACCAG | | |
| blaOXA-48 | Oxa-$_{48}$(+) | GCGTGGTTAAGGATGAACAC | | |
| | Oxa-$_{48}$(-) | CATCAAGTTCAACCCAACCG | | |

## Statistical analysis

Potential risk factors associated with *ESBL-KP* colonization were studied. Statistical analysis was performed with SPSS version 26, where additional variables were created for the analysis. Frequencies (percentages) of qualitative variables and mean values (standard deviation) of continuous were calculated. The chi-square test for continuous variables and Fisher's exact test for categorical and nominal variables were used to make variable comparisons for cases with versus without colonization. Logistic regression models were used to identify variables related to ESBL-KP acquisition events. Multivariate adjusted odds ratios (ORs) and corresponding 95% confidence intervals (CIs) were estimated and adjusted for: Admission age (day), gender (male, female), birth Weight (g) ($< 2500$ g, $\geq 2500$ g), delivery mode (vaginal birth, cesarean section), invasive procedure (endotracheal tube, enteral tube feeding), antimicrobial exposure 3rd CG, feeding (breast milk, milk formula) and PPROM History (no PPROM, PPROM$> 18$ hours). All analyses were 2-tailed, and a p-level of $< 0.05$ was considered statistically significant.

# Results

## Population characteristics

One hundred and thirteen preterm infants were prospectively recruited for one year. Male neonates made up the majority of their 66 study participants (58.40%). The majority of 98 (86.72%) premature newborns had a birth weight of less than 2500 g and a median of 1736.12 g (IQR: 1400–2150). 83 (73.45%) of the newborns were delivered by cesarean section. 62 (54.86%) were born very preterm (gestational age: 28 to $<32$ weeks) and 87 (76.99%) with a respiratory distress complication "Table 2".

## Microbiological results

Of the 113 preterm newborn infants included in the study, 62 (54.8%) premature infants acquired *ESBL-KP*. The molecular and antibiotic susceptibility data of *ESBL-producing K. Pneumoniae* isolates are shown in "Table 3". We found that *ESBLs K. Pneumoniae* isolates were 100% resistant to *amoxicillin, clavulanic acid-amoxicillin, cefotaxime, ceftazidime, and gentamicin*. The most significant resistance was observed in the *TEM β-lactamases* category to *nalidixic acid, norfloxacin, and/or ciprofloxacin, sulfamethoxazole* with (84.6%) followed by the *CTX-M-2 β-lactamases* category (60%). The resistance pattern of the *SHV β-lactamases K. Pneumoniae* isolates to *nalidixic acid, norfloxacin, and/or ciprofloxacin, sulfamethoxazole, ertapenem* was 46.1%,43.5%,46.1%, and 2.5%, respectively and two strains characterized as *OXA-48* showed resistance to ertapenem. However, the data indicated that all *ESBL* isolates from *K. Pneumoniae* showed susceptibility to *amikacin*.

## Risk factors analyses

The bivariate analysis suggested that type of delivery, previous antibiotic use, *Invasive procedures*, and feeding were significantly associated with the acquisition event, P $< 0.05$; "Table 2" ». Out of the 83 neonates with Caesarean delivery mode who have been admitted to the neonatal intensive care unit (NICU), 50(80.64%) acquired *ESBL-KP* compared to 12(19.35%) of those with Vaginal delivery mode (P = 0.045). It was also noted that neonates born to mothers without *PPROM* history were more likely to be colonized with *ESBL-KP* than those born to mothers with a *PPROM* history of more than 18 hours 51(82.25%) vs. 11(17.74%), P = 0.391. For the invasive procedure, the preterm neonates with an endotracheal tube and an enteral tube feeding probe have a high *ESBLs K. pneumoniae* colonization rate of 45 (72.58%)–60

**Table 2. Characteristics and risk factors data of preterm newborns infants according to the acquisition of *ESBLs producing K. Pneumoniae*.**

| Characteristics | All preterm N = 113 | Neonates with *ESBLs* producing *K. Pneumoniae* acquisition | | p-value |
|---|---|---|---|---|
| | N. (%) | Yes N. (%) | No N. (%) | |
| Age at admission (Day)* | IQR: 01–01 | 1.70 ±2.09 | 5.70 ±15.58 | 0.048 |
| Gender | | | | |
| • Male | 66(58.40) | 32(51.61) | 34 (66.66) | 0.076 |
| • Female | 47(41.59) | 30(48.38) | 17(33.33) | |
| Birth Weight (g) Mean* | IQR:1400–2150 | 1736.12 ±451.72 | 1884.70 ±610.76 | 0.140 |
| • ≥ 2500 g | 15(13.27) | 05 (8.06) | 10 (19.60) | 0.064 |
| • < 2500 g | 98(86.72) | 57 (91.93) | 41 (80.39) | |
| PPROM History | | | | |
| • No PPROM | 91(80.53) | 51(82.25) | 40(78.43) | 0.391 |
| • PPROM> 18 hours | 22(19.46) | 11(17.74) | 11(21.56) | |
| Delivery Mode | | | | |
| • Vaginal birth | 30(26.54) | 12(19.35) | 18(35.29) | 0.045 |
| • Caesarean birth | 83(73.45) | 50(80.64) | 33(64.70) | |
| Prematurity | | | | |
| • Moderate and late preterm (32 to 37 weeks) | 28(24.77) | 20(32.25) | 08 (15.68) | 0.125 |
| • Very preterm (28 to <32 weeks) | 62(54.86) | 31(50.00) | 31(60.78) | |
| • Extremely preterm (less than 28 weeks) | 23(20.35) | 11(17.74) | 12(23.52) | |
| Associated pathology | | | | 0.850 |
| • Congenital Defect and/or Surgical Pathology | 12(10.61) | 06(9.67) | 06(11.76) | |
| • Neonatal suffering | 14(12.38) | 07(11.29) | 07(13.72) | |
| • Respiratory Distress | 87(76.99) | 49(79.03) | 38(74.50) | |
| Length of stay (Day)* | IQR:05–11 | 8.01±4.25 | 10.03±7.95 | 0.087 |
| • <5 Day | 20(17.69) | 11(17.74) | 09(17.64) | 0.988 |
| • 5–10 Day | 59(52.21) | 32(51.61) | 27(52.94) | |
| • >10 Day | 34(30.08) | 19(30.64) | 15(29.41) | |
| Invasive procedures | | | | |
| • PVC | 110(97.34) | 61(98.38) | 49(96.07) | 0.426 |
| • Endotracheal tube | 63(55.75) | 45(72.58) | 18(35.29) | <0.001 |
| • UMCAT | 43(38.05) | 24(38.70) | 19(37.25) | 0.515 |
| • CPAP | 104(92.03) | 55(88.70) | 49(96.07) | 0.137 |
| • Enteral tube feeding | 77(68.14) | 60(96.77) | 17(33.33) | <0.001 |
| Antimicrobial exposure | | | | |
| • Aminopenicillin | 104(92.03) | 57(91.93) | 47(92.15) | 0.622 |
| • 3rd CG | 58(51.32) | 38(65.51) | 20(34.48) | 0.023 |
| • Aminoglycoside | 105(92.92) | 58(93.54) | 47(92.15) | 0.527 |
| • Quinolone and /or Imipenem | 12(10.61) | 05(8.06) | 07(13.72) | 0.255 |
| Feeding | | | | |
| • Milk formulae | 29(25.66) | 22(35.48) | 07(13.72) | 0.006 |
| • Breast milk | 84(74.33) | 40(64.51) | 44(86.27) | |

UMCAT: Umbilical Catheter, CPAP: Continuous Positive Airway Pressure, PVC: Peripheral Venous Catheter, PPROM: Prolonged Premature Rupture of Membrane, 3rd CG: Third Cephalosporine Generation.

*IQR, Mean ± SEM.

**Table 3. Molecular and Antibiotic sensitivity data of acquired *ESBL producing K. Pneumoniae* isolates.**

| Gene (s) detected by PCR | N (%) | Antibiotic sensitivity pattern of ESBL K. Pneumoniae isolates N = 62 (54,8%) | | | | | | | | | |
|---|---|---|---|---|---|---|---|---|---|---|---|
| | | AMX (10 μg) | AMC (20/10 μg) | CTX (30 μg) | CAZ (30 μg) | GEN (10 μg) | AMK (30 μg) | NAL (30 μg) | NOR and/or CIP 10 μg /5 μg | SXT 1.25/ 23.75 μg | ETP (10 μg) |
| SHV β-lactamases | 51 (82) | 51 (100%) | 51 (100%) | 51 (100%) | 51 (100%) | 51 (100%) | - | 24(46.1%) | 23(43.5% | 24(46.1%) | 2(2.5%) |
| TEM β-lactamases | 30 (48) | 30 (100%) | 30 (100%) | 30 (100%) | 30 (100%) | 30 (100%) | - | 26(84.6%) | 26(84.6%) | 26(84.6%) | 2(3.8%) |
| CTX-M$_{-1}$ β-lactamases | 58 (93.5) | 58 (100%) | 58 (100%) | 58 (100%) | 58 (100%) | 58 (100%) | - | 33(56.6%) | 32(54.7%) | 33(56.6% | 2(1.8%) |
| CTX-M$_{-2}$ β-lactamases | 03 (4.8) | 03 (100%) | 03 (100%) | 03 (100%) | 03 (100%) | 03 (100%) | - | 2(60%) | 2(60%) | 2(60%) | - |
| CTX-M$_{-9}$ β-lactamases | 07(11.2) | 07 (100%) | 07 (100%) | 07 (100%) | 07 (100%) | 07 (100%) | - | 4(44.4%) | 4(44.4%) | 4(44.4%) | - |
| OXA-48 β-lactamases | 02(3.22) | 02 (100%) | 02 (100%) | 02 (100%) | 02 (100%) | 02 (100%) | - | 01 (50%) | 02 (100%) | 02 (100%) | 02 (100%) |

(96.77%), P = 0.001, respectively. Preterm neonates exposed to third-generation cephalosporin antibiotics have acquired *ESBL-KP* at a lower rate than those who did not, 38 (65.51%) vs. 58 (51.32%), P = 0.023 "Table 2". All the variables analyzed in the bivariate analysis were associated with the risk of colonization (P < 0.25) and were therefore examined in the multiple logistic regression analysis.

The multivariate regression analysis reveals that premature infants with enteral tube feeding and endotracheal tubes have a significantly higher risk of acquiring *ESBL-KP* (p value = 0.001). Premature infants with caesarean section mode had a higher risk compared to premature with vaginal birth (OR = 2.27, 95% CI: 0.96–5.33, p-Value = 0.059). Among the antimicrobial risk factors, premature infants who were exposed to third-generation cephalosporins during hospitalization had a higher risk of acquiring *ESBL-KP* compared with preterm infants who were not exposed (OR = 2. 45, 95% CI: 1.14–5.24, p-Value = 0.020), and a moderate risk was observed in preterm infants who received milk formulae compared with those who received breast milk (OR = 0.28, 95% CI: 0.11–0.74, p-Value = 0.010). After adjustment for potential confounders, "Table 4", premature neonates who received enteral tube feeding had a significantly higher risk of *ESBL-KP* acquisition than neonates who did not receive enteral tube feeding (OR = 38.46, 95% CI: 7.86–188.20, p-value < 0.001). Compared to neonates without endotracheal tubes, premature babies with endotracheal tubes had a significantly higher risk of *ESBL-KP* acquisition (OR = 4.86, 95% CI: 1.37–17.19, p-value <0.014).

There was no significant association in the multivariate analysis with *ESBL-KP* acquisition risk for all prematurity levels, Age at admission, gender, feeding, birth weight (< 2500 g, ≥ 2500 g), Delivery Mode, and antimicrobial exposure.

## Discussion

*K. pneumoniae* infections are particularly *troublesome* among hospitalized newborns [23]. This strain is the most common cause of sepsis and epidemics in perinatal intensive care units [1,24]. Several reviews have highlighted that hospitalized patients carrying *K. pneumoniae* at the intestinal level developed fulminant infections with the same strain of carriage [25,26]. For one year, we investigated the molecular characterization of acquired *K. Pneumoniae* in the intestinal tract of hospitalized premature infants. Among 113 preterm infants included in this study, 62 (54.8%) acquired *ESBL-KP*. Premature infants are known to have an abnormal gut colonization pattern in the first few weeks of life [27,28], which may lead to an increased susceptibility to disease [29,30]. Compared to full-term infants, the gut microbiota of preterm infants has significantly reduced bacterial diversity and an abundance of microorganisms typically associated with hospital environments [31,32]. In this study the resistance profile genes

**Table 4. Logistic regression models of potential risk factors predicting *ESBL-producing K. Pneumoniae* acquisition in the intestinal tract during hospitalization.**

| Risk factors | | Unadjusted Odds Ratio | | Adjusted Odds Ratio [a] | |
|---|---|---|---|---|---|
| | | OR (95% CI) | *p-Value* | OR (95% CI) | *p-Value* |
| Age at admission (Day) | | 0.94(0.86–1.02) | 0.154 | 0.94 (0.85–1.04) | 0.301 |
| Gender | | | | | |
| • Male | | 1 | | 1 | |
| • Female | | 1.87(0.87–4.03) | 0.107 | 1.30 (0.41–4.10) | 0.651 |
| Birth Weight (g) Mean continuous | | 0.99 (0.99–1.00) | 0.142 | 1.00(0.99–1.00) | 0.109 |
| • ≥ 2500 g | | 1 | | 1 | |
| • < 2500 g | | 2.78 (0.88–8.74) | 0.080 | 0.67(0.10–4.14) | 0.067 |
| Delivery Mode | | | | | |
| • Vaginal birth | | 1 | | 1 | |
| • Caesarean birth | | 2.27 (0.96–5.33) | 0.059 | 3.55(0.88–14.22) | 0.072 |
| Prematurity | | | | | |
| • Moderate and late preterm (32 to 37 weeks) | | 1 | | 1 | |
| • Very preterm (28 to <32 weeks) | | 0.40(0.15–1.04) | 0.061 | 0.83(0.21–3.33) | 0.802 |
| • Extremely preterm (less than 28 weeks) | | 0.36 (0.11–1.16) | 0.089 | 1.20 (0.21–6.64) | 0.831 |
| Invasive procedures | | | | | |
| • PVC | No | 1 | | 1 | |
| | Yes | 2.48 (0.21–28.27) | 0.461 | 6.23 (0.28–135.43) | 0.243 |
| • Endotracheal tube | No | 1 | | 1 | |
| | Yes | 4.85 (2.17–10.8) | <0.001 | 4.86 (1.37–17.19) | 0.014 |
| • UMCAT | No | 1 | | 1 | |
| | Yes | 1.06 (0.49–2.28) | 0.874 | 2.65 (0.73–9.60) | 0.137 |
| • CPAP | No | 1 | | 1 | |
| | Yes | 0.32 (0.06–1.61) | 0.168 | 0.21 (0.016–2.97) | 0.254 |
| • Enteral tube feeding | No | 1 | | 1 | |
| | Yes | 59.9 (13.0–275.5) | <0.001 | 38.46(7.86–188.20) | <0.001 |
| Antimicrobial exposure | | | | | |
| • Aminopenicillin | No | 1 | | 1 | |
| | Yes | 0.97 (0.24–3.81) | 0.965 | 2.51 (0.46–13.4) | 0.281 |
| • 3rd CG | No | 1 | | 1 | |
| | Yes | 2.45 (1.14–5.24) | 0.020 | 1.48 (0.48–4.49) | 0.487 |
| • Aminoglycoside | No | 1 | | 1 | |
| | Yes | 1.23 (0.29–5.19) | 0.774 | 0.26 (0.04–1.57) | 0.145 |
| • Quinolone and /or Imipenem | No | 1 | | 1 | |
| | Yes | 0.55 (0.16–1.85) | 0.336 | 0.38(0.06–2.19) | 0.280 |
| Feeding | | | | | |
| • Milk formulae | | 1 | | 1 | |
| • Breast milk | | 0.28 (0.11–0.74) | 0.010 | 0.43(0.11–1.63) | 0.218 |

OR: Odds ratio; CI: Confidence interval.

[a]Odds ratios adjusted for Admission Age (Day), Delivery Mode (Vaginal birth, Caesarean birth), Invasive procedure (Endotracheal tube, Enteral tube feeding), Antimicrobial exposure 3rd CG, Feeding (Breast milk, Milk formulae), PPROM History (No PPROM, PPROM> 18 hours).

result of *ESBL-KP* are presented below: SHV β-lactamases 51 (82%), *TEM β-lactamases* 30 (48%), *CTX-M-1 β-lactamases* 58 (93.5%), *CTX-M-2 β-lactamases* 03 (4.8%), *CTX-M-9 β-lactamases* 07(11.2%) and *OXA-48 β-lactamases* 02 (3.22%), This high degree of prevalence has been frequently reported in investigations focused on gut colonization [1,33]. Still, this

molecular diversity is not surprising since opportunistic microorganisms *K. pneumoniae* are known to acquire resistance genes through antibiotic pressure [34].

To our knowledge, this is the first study in our country that identified endotracheal intubation and enteral tube feeding as independent risk factors for colonization of neonates with *ESBL-KP* both were significant in the multiple regression analysis (p-value = 0,001). Invasive procedures are usually associated with *ESBL-KP* colonization and/or infection among hospitalized neonates [35]. Other studies reported that the method of feeding, as well as the nature of enteral feeds, are important factors in early gut colonization [16,36]. Petersen et al. have shown that enteral tube feeding resulted in a high bacterial density on the first day of use [37]. Also, several studies show that the bacterial flora in the feeding tubes of newborns can influence the bacterial colonization of the intestinal tract [12,38].

However, Crivaro et al. [39] and Cassettari et al. [40] observed that empirical antibiotic treatment is a significant risk factor associated with *ESBL-KP* employment status in newborns. Our results confirmed this, in the univariate analysis and unadjusted regression, especially for the *third generation of cephalosporins* exposed under ESBL-KP acquisition. By performing a multi-logistic regression analysis, we found limited associations with early empirical antibiotic therapy. It seems that not antibacterial therapy itself but poor hospital hygiene enables the circulation and transmission of multiresistant strains, which then requires broad-spectrum antibacterial agents and results in longer NICU length of stay and greater potential for colonization by resistant organisms [41,42].

Previous studies have also emphasized the role of both anamnestic and infant-related risk factors, such as *PPROM*, vaginal birth, birth weight, preterm birth rate, and length of stay, for the acquisition of *ESBL-KP* [17–19]. These findings were not seen in the present study and could presumably be attributed to the small number of cases.

Moreover, it seems that *ESBL Enteroacteriacae* colonized mothers are an independent risk factor for the colonization of neonates with *ESBL-Enterobacteria*. She might harbor these bacteria in their normal intestinal flora and contaminate their newborns during birth. To decrease neonatal morbidity and mortality, several studies suggested systematic screening of the intestinal flora of premature newborns and their mothers, should be implemented in neonatal wards [2,19].

Limitations of our study include the small sample size; therefore, inference about causality is limited. in addition, it is unusual that a study could identify all the risk factors associated with the acquisition of these multi-resistant bacteria that play an essential role in causing infection in NICUs. The difficulty of the subject is due to the complexity of tracking newborns after birth. Many factors limit studies in this regard, including operating costs, deaths, short hospitalization times, and the difficulty of following anamnestic and biological clinical data [33,43].

## Conclusion

The acquisition of *ESBL-KP* carriage was demonstrated in 54.8% of preterm infants. Our finding supposes that enteral tube feeding and endotracheal tubes may be independent risk factors for colonization in the intestinal tract with *ESBL-KP* during a hospital stay in the neonatal intensive care unit. This highlights the current status of two practices that will require procedural updating in the NICU.

## Supporting information

**S1 File.**
(XLSX)

## Author Contributions

**Conceptualization:** Benboubker Moussa, Bouchra Oumokhtar, Samira Elfakir, Fouzia Hmami.

**Data curation:** Samira Elfakir.

**Formal analysis:** Benboubker Moussa, Bouchra Oumokhtar, Btissam Arhoune, Abdelhamid Massik, Hammad Soudi.

**Investigation:** Benboubker Moussa, Bouchra Oumokhtar, Abdelhamid Massik, Hammad Soudi, Fouzia Hmami.

**Methodology:** Benboubker Moussa, Btissam Arhoune, Abdelhamid Massik, Fouzia Hmami.

**Project administration:** Fouzia Hmami.

**Resources:** Benboubker Moussa.

**Software:** Samira Elfakir, Mohamed Khalis.

**Supervision:** Bouchra Oumokhtar, Btissam Arhoune, Fouzia Hmami.

**Validation:** Btissam Arhoune, Abdelhamid Massik, Mohamed Khalis.

**Visualization:** Benboubker Moussa, Samira Elfakir, Mohamed Khalis.

**Writing – original draft:** Benboubker Moussa, Mohamed Khalis, Hammad Soudi, Fouzia Hmami.

**Writing – review & editing:** Samira Elfakir, Fouzia Hmami.

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
