## [Decision Letter · Decision Letter 0]

1 Dec 2022

PONE-D-22-25668Gut acquisition of Extended-spectrum β-lactamases-producing Klebsiella pneumoniae in preterm neonates:PLOS ONE

Dear Dr. moussa,

Thank you for submitting your manuscript to PLOS ONE. After careful consideration, we feel that it has merit but does not fully meet PLOS ONE’s publication criteria as it currently stands. Therefore, we invite you to submit a revised version of the manuscript that addresses the points raised during the review process.

We look forward to receiving your revised manuscript.

Kind regards,

Grzegorz Woźniakowski, Full professor, PhD, ScD

Academic Editor

PLOS ONE

Journal Requirements:

2. "PLOS requires an ORCID iD for the corresponding author in Editorial Manager on papers submitted after December 6th, 2016. Please ensure that you have an ORCID iD and that it is validated in Editorial Manager. To do this, go to ‘Update my Information’ (in the upper left-hand corner of the main menu), and click on the Fetch/Validate link next to the ORCID field. This will take you to the ORCID site and allow you to create a new iD or authenticate a pre-existing iD in Editorial Manager. Please see the following video for instructions on linking an ORCID iD to your Editorial Manager account: " ext-link-type="uri" xlink:type="simple">https://www.youtube.com/watch?v=_xcclfuvtxQ"

3. Please provide additional details regarding participant consent. In the ethics statement in the Methods and online submission information, please ensure that you have specified what type you obtained (for instance, written or verbal, and if verbal, how it was documented and witnessed). If your study included minors, state whether you obtained consent from parents or guardians. If the need for consent was waived by the ethics committee, please include this information

Reviewers' comments:

Reviewer's Responses to Questions

**Comments to the Author**

1. Is the manuscript technically sound, and do the data support the conclusions?

Reviewer #1: Yes

2. Has the statistical analysis been performed appropriately and rigorously? 

Reviewer #1: Yes

3. Have the authors made all data underlying the findings in their manuscript fully available?

Reviewer #1: Yes

4. Is the manuscript presented in an intelligible fashion and written in standard English?

Reviewer #1: Yes

5. Review Comments to the Author

Reviewer #1: This manuscript describes investigations of ESBL producing Klebsiella pneumoniae strains obtained from preterm babies. In my opinion topic of this manuscript is important however, further investigations are required to strenghten statements of this study.

1) In this study ESBL producing strains were analyzed but 2 strains were ertapenem resistant. What resistance determinant caused this phenotype ? PCR for NDM, KPC, VIM should be performed.

2) What was the clonal relatedness of ESBL producing K. pneumoniae in this study? Pulsed-field gelectrophoresis or multilocus sequence typing should be performed.

3) There are some phrasis in the text that are not properly used.

In absract: "Klebsiella spp. are commonly enriched in the intestinal microbiota..:" proper form: Klebsiella spp. can colonise the intestine of preterm neonates...

In introduction: "Fact, Klebsiella is a reservoir for antibiotic-resistant genes that can spread to other gram-negative bacteria..

Proper form: Antibiotic resistance genes can be exchanged between gram-negative bacteria through horizontal gene transfer...

6. PLOS authors have the option to publish the peer review history of their article (what does this mean?). If published, this will include your full peer review and any attached files.

Reviewer #1: No

---

## [Author Response · Author response to Decision Letter 0]

21 Jan 2023

PLOS ONE's style requirements: The manuscript has been checked and corrections have been made as recommended by the journal

ORCID iD for the corresponding author: The ORCID iD for the corresponding author has been added

Additional details regarding participant consent:Additional details regarding participant consent have been added in the Ethical consideration chapter: informed consent was obtained from the parents according to a written format. Parents may remove their consent at any time during the study.

Financial disclosure and funding:A new chapter has been added mentioning that the authors received no specific funding for this work

Reviewer #1: This manuscript describes investigations of ESBL producing Klebsiella pneumoniae strains obtained from preterm babies. In my opinion topic of this manuscript is important however, further investigations are required to strenghten statements of this study.

1. In this study, ESBL producing strains were analyzed but 2 strains were ertapenem resistant. What resistance determinant caused this phenotype? PCR for NDM, KPC, VIM should be performed:Indeed , both strains have been characterized already after the submission ,the results showed that they encode to OXA-48 genes. For the other genes KPC, VIM and NDM the results were negative these results have been added to the manuscript

2. What was the clonal relatedness of ESBL-producing K. pneumoniae in this study? Pulsed-field electrophoresis or multilocus sequence typing should be performed: In this work, as you know, the aim is to study only the prevalence and factors associated with intestinal colonization of preterm infants with KP-ESBL.We rely on the PCR technique on agarose gel for a possible genetic characterization using the recommended primers.At the moment and due to lack of funding. We do not have other techniques in our laboratory to study the recommended clonal relationships on the strains investigated in this work such as PFGE and MLST.

Please note that our global project foresees in perspective the study of the relationship between KP-ESBL strains from intestinal carriage and strains isolated after the occurrence of bacteremia in our study population after having the funding.

3. There are some phrasis in the text that are not properly used.

In absract: "Klebsiella spp. are commonly enriched in the intestinal microbiota.:" proper form: Klebsiella spp. can colonise the intestine of preterm neonates.:The sentence has been replaced by the recommended sentence in the abstract: Klebsiella spp. can colonize the intestine of preterm neonates

4. in introduction: "Fact, Klebsiella is a reservoir for antibiotic-resistant genes that can spread to other gram-negative bacteria..Proper form: Antibiotic resistance genes can be exchanged between gram-negative bacteria through horizontal gene transfer..: The sentence has been replaced by the recommended in the introduction: Fact, Antibiotic resistance genes can be exchanged between gram-negative bacteria through horizontal gene transfer

---

## [Decision Letter · Decision Letter 1]

15 Jun 2023

PONE-D-22-25668R1Gut acquisition of Extended-spectrum β-lactamases-producing Klebsiella pneumoniae in preterm neonates:  critical role of enteral feeding, and endotracheal tubes in the neonatal intensive care unit (NICU)PLOS ONE

Dear Dr. moussa,

Thank you for submitting your manuscript to PLOS ONE. After careful consideration, we feel that it has merit but does not fully meet PLOS ONE’s publication criteria as it currently stands. Therefore, we invite you to submit a revised version of the manuscript that addresses the points raised during the review process.

We look forward to receiving your revised manuscript.

Kind regards,

Grzegorz Woźniakowski, Full professor, PhD, ScD

Academic Editor

PLOS ONE

Journal Requirements:

Reviewers' comments:

Reviewer's Responses to Questions

**Comments to the Author**

1. If the authors have adequately addressed your comments raised in a previous round of review and you feel that this manuscript is now acceptable for publication, you may indicate that here to bypass the “Comments to the Author” section, enter your conflict of interest statement in the “Confidential to Editor” section, and submit your "Accept" recommendation.

Reviewer #2: All comments have been addressed

2. Is the manuscript technically sound, and do the data support the conclusions?

Reviewer #2: Yes

3. Has the statistical analysis been performed appropriately and rigorously? 

Reviewer #2: Yes

4. Have the authors made all data underlying the findings in their manuscript fully available?

Reviewer #2: Yes

5. Is the manuscript presented in an intelligible fashion and written in standard English?

Reviewer #2: Yes

6. Review Comments to the Author

Reviewer #2: The article if interesting. There are only small errors which should be corrected.

Bellowed comments refers to the “Revised Manuscript with Track Changes” version.

Lines 32, 256: “K. Pneumoniae” should be written in italic.

Line 54: There is unnecessary dot after the word “Klebsiella”

Reference list has a some gaps such us year of article in point 4 and 7. It should list should be checked carefully.

However in my opinion the article should be published after minor revision.

Sincerely,

Reviewer

7. PLOS authors have the option to publish the peer review history of their article (what does this mean?). If published, this will include your full peer review and any attached files.

Reviewer #2: No

---

## [Author Response · Author response to Decision Letter 1]

4 Jul 2023

Lines 32, 256: “K. Pneumoniae” should be written in italic :"K. Pneumoniae" has been converted to italics

Line 54: There is unnecessary dot after the word “Klebsiella” : Dot has been deleted

Reference list has a some gaps such us year of article in point 4 and 7. It should list should be checked carefully. Both existing references are not correctly displayed on the list with the style recommended by the journal automaticully, so we have replaced references 4 and 7 with other references that begin in the same context in the manuscript and are presented as follows:

Reference 4 [Cotton MF, Wasserman E, Pieper CH, Theron DC, van Tubbergh D, Campbell G, et al. Invasive disease due to extended spectrum beta-lactamase-producing Klebsiella pneumoniae in a neonatal unit:the possible role of cockroaches. :5.] has been replaced by [Pavez M, Troncoso C, Osses I, Salazar R, Illesca V, Reydet P, et al. High prevalence of CTX-M-1 group in ESBL-producing enterobacteriaceae infection in intensive care units in southern Chile. Braz J Infect Dis. 2019 Apr 24;23(2):102–10.]

And reference 7 [Sakai AM, Iensue TNAN, Pereira KO, Silva RL da, Pegoraro LG de O, Salvador MS de A, et al. Colonization profile and duration by multi-resistant organisms in a prospective cohort of newborns after hospital discharge. Rev Inst Med Trop São Paulo. 2020;62:e22.] has been remplaced by[Milic M, Siljic M, Cirkovic V, Jovicevic M, Perovic V, Markovic M, et al. Colonization with Multidrug-Resistant Bacteria in the First Week of Life among Hospitalized Preterm Neonates in Serbia: Risk Factors and Outcomes. Microorganisms. 2021 Dec 17;9(12):2613.]

---

## [Decision Letter · Decision Letter 2]

24 Oct 2023

Gut acquisition of Extended-spectrum β-lactamases-producing Klebsiella pneumoniae in preterm neonates:  critical role of enteral feeding, and endotracheal tubes in the neonatal intensive care unit (NICU)

PONE-D-22-25668R2

Dear Dr. moussa,

We’re pleased to inform you that your manuscript has been judged scientifically suitable for publication and will be formally accepted for publication once it meets all outstanding technical requirements.

Kind regards,

Monica Cartelle Gestal, PhD

Academic Editor

PLOS ONE

Additional Editor Comments (optional):

Reviewers' comments:

Reviewer's Responses to Questions

**Comments to the Author**

1. If the authors have adequately addressed your comments raised in a previous round of review and you feel that this manuscript is now acceptable for publication, you may indicate that here to bypass the “Comments to the Author” section, enter your conflict of interest statement in the “Confidential to Editor” section, and submit your "Accept" recommendation.

Reviewer #2: All comments have been addressed

2. Is the manuscript technically sound, and do the data support the conclusions?

Reviewer #2: (No Response)

3. Has the statistical analysis been performed appropriately and rigorously? 

Reviewer #2: (No Response)

4. Have the authors made all data underlying the findings in their manuscript fully available?

Reviewer #2: (No Response)

5. Is the manuscript presented in an intelligible fashion and written in standard English?

Reviewer #2: (No Response)

6. Review Comments to the Author

Reviewer #2: (No Response)

7. PLOS authors have the option to publish the peer review history of their article (what does this mean?). If published, this will include your full peer review and any attached files.

Reviewer #2: No

---

## [Editor Report · Acceptance letter]

27 Oct 2023

PONE-D-22-25668R2 

Gut acquisition of *Extended-spectrum β-lactamases-producing Klebsiella pneumoniae* in preterm neonates:  critical role of enteral feeding, and endotracheal tubes in the neonatal intensive care unit (NICU) 

Dear Dr. Moussa:

I'm pleased to inform you that your manuscript has been deemed suitable for publication in PLOS ONE. Congratulations! Your manuscript is now with our production department. 

Kind regards, 

on behalf of

Dr. Monica Cartelle Gestal 

Academic Editor

PLOS ONE